# Microbial Resistance to Carbapenems in Effluents from Gynaecological, Paediatric and Surgical Hospital Units

**DOI:** 10.3390/antibiotics11081103

**Published:** 2022-08-15

**Authors:** El hassan Loumame, Abdessamad Tounsi, Soumia Amir, Nabila Soraa, Naaila Ouazzani

**Affiliations:** 1Environmental, Ecological and Agro-Industrial Engineering Laboratory, University Sultan Moulay Slimane, Beni-Mellal 23000, Morocco; 2Microbiology Laboratory, University Hospital Mohamed VI, Cadi Ayyad University, Marrakech 40000, Morocco; 3Laboratory of Water, Biodiversity and Climate Change, Faculty of Sciences Semlalia, Cadi Ayyad University, Marrakech 40000, Morocco; 4Centre for Agrobiotechnology and Bioengineering, Cadi Ayyad University, Marrakech 40000, Morocco

**Keywords:** antibiotic-resistance, hospital effluents, carbapenems

## Abstract

The aim of this work is to identify and count antimicrobial resistance (AMR) in hospital effluents (HEs) of 2 units of the University Hospital Mohamed VI the Mother and Child Hospital (MCH) and the Ar-razi Surgical Hospital (ArzH), and to compare the two hospital units in terms of ARMs and seasonal variation. Each HE was sampled during 2016 and 2017. After identification of the pathogenic strains and determination of AMR, the results were reported for 24 ABs, including 3 carbapenems (CBP), and their consumption rates. The Predicted environmental concentration (PEC) rate of carbapenems in the HE of the study sites is calculated. A comparative analysis of the AMR of the isolated bacterial species was performed and related to the evolution of PEC in HEs. In the ArzH effluents:15 strains isolated, 7 are carbanepenem-resistant Enterobacteria (CRE) and are resistant to at least one of the 3 carbapenems tested. ArzH and MCH effluents respectively show some similarities: 26.87% and 28.57% of isolated bacteria are resistant to ertapenem while 43.48% and 57.14% are resistant to meropenem. However, for imipenem, the MCH effluent has a higher percentage of bacterial antibiotic resistance than ArzH. In addition, the percentage of resistance in each hospital unit effluent is mainly in relation with the increasing antibiotic consumption and predicted environmental values PEC for very antibiotic in each unit in the same period.

## 1. Introduction

Antibiotics have saved millions of lives since their discovery and application in the treatment of human and animal bacterial infections. However, their widespread use has led to the emergence and spread of antibiotic resistance among pathogenic bacteria, which poses a risk to human health [1]. Global public health is therefore threatened by antibiotic-resistant bacteria, which are increasing in number and rendering antibiotics ineffective or at least diminishing their effect [2].

This major threat of antibiotic resistance leads to increased mortality and longer hospital stays for patients at higher costs [2]. Beta-lactams are by far the most widely used antibiotics in the world, including penicillin, cephalosporins, monobactams and carbapenems. These antibiotics all contain a beta-lactam ring and have the ability to inhibit cell wall synthesis of a wide range of pathogens.

Carbapenems, among the beta-lactam antibiotics, are the most effective against Gram-positive and Gram-negative bacteria, with a broad spectrum of antibacterial activity. In addition, Carbapenems are often antibiotics of last resort used to treat multidrug-resistant infections [3]. Futhermore, carbapenems have fewer side effects and are safer to use than other drugs of last resort such as polymyxins [4]. In fact, the World Health Organization(WHO), in its latest review (WHO report 6th revision 2018) ranked the carbapenem family third out of 18: even more important than the penicillin family, the 3rd, 4th, and 5th generation cephalosporins; macrolides, polymyxins, and quinolones, and other antibiotics.

Many of the most dangerous antibiotic-resistant germ infections are caused by CRE, as carbapenems are potent antibiotics and the most effective against multidrug-resistant bacterial infections [5,6] and indeed the most widely used. In the 2013 classification, the Centers for Disease Control and Prevention (CDC) considered the threat of CRE, among the top three "urgent" threats [7].

The first epidemiologic report on the emergence of CRE, was made by Kumarasamy et al. [4] in India, Pakistan, and the United Kingdom [8]. Since then, more than 975 articles related to carbapenemases, their mechanisms of action, epidemiology and treatment have been indexed in PubMed [9]. CRE, have been identified in recent years as one of the major threats to human health [10]. The most dangerous CREs are known to produce the enzyme carbapenemase [10]. These bacteria with antimicrobial resistance to carbapenems can hydrolyze these ABs and also break down other beta-lactam antibiotics [11]. CREstrains are also enteric beta-lactamase bacteria that typically produce carbapenemases, which are plasmid-carried enzymes that can be readily shared with other gram-negative bacteria [12]. These carbapenemases constitute a diverse group of β-lactamases belonging to ambler classes A, B, and D that deactivate carbapenems [11].

Genes encoding carbapenem deactivating enzymes have an efficient dissemination, subsequently, they will have a wide dissemination in the bacterial world [8]. In addition, additional resistances have been co-transferred, indicating a co-localization of antibiotic resistance genes on plasmids, which makes them more mobile [8].

In Morocco, surveys conducted between 2005 and 2010 on nosocomial infections (NI) in different university hospitals showed a very large use of AB: 20% and 56.4%, 32.8% and 42.9% respectively in Rabat and Fez in 2005 and 2010 in addition to a high prevalence of NI (in 2005 17.8% in Ibn Sina Hospital in Rabat, 8.2% in Fez University Hospital, and in 2010: 10.3% in Rabat and 13.3% in Fez). In the different surveys, very high rates of bacterial resistance were noted: 50% of *Staphylococcus* spp. resistant to methicillin, 45.5% of *Klebsiella* spp. Extended-spectrum producing beta-lactamases (ESBL) and 66.7% of Acinetobacter resistant to Imipenem [12].

This resistance is probably due, among other things, to the increase in consumption levels of these antibiotics. In France, carbapenem consumption increased rapidly between 2012 and 2016, from 0.021 to 0.033 Defined Daily Doses (DDD)/1000 inhabitants per day [8].

All of these studies have reported antibiotic-resistant bacterial strains isolated from patients inside hospitals. A few studies have characterized the level of antibiotic resistance of strains in the environment, particularly in hospital wastewater [13]. Nevertheless, several studies have pointed out that in some cases, 60% of the pollutant loads entering wastewater treatment plants come from hospital effluent [14,15]. In addition, levels of antibiotic resistant genes were highest in hospital HEs [14].

In the effluent of a Brazilian university hospital and at different wastewater sewage plant sites, most isolates had the blaK PC-2 gene hosted on a transposon that was carried by conjugate plasmids [16]. In the same study, the presence of *Klebsiella pneumoniae* carbapenemase production by *Aeromonas* spp., *Kluyvera* spp., and *Enterobacteriaceae* indicated the adaptability of these isolates to aquatic environments, not only in the hospital effluent but throughout the wastewater treatment plant. According to the same authors, among the bacteria with the Imipenem resistance gene, several representatives of the genera *Klebsiella* spp. (41.9%), *Enterobacter* spp. (24.2%), *Citrobacter* spp. (8.1%), *Serratia* spp. (4.8%) and *Raoultella* spp. (6.4%), *Aeromonas* spp. (6.4%) *and Kluyvera* spp. (8.1%) were detected [16]. In addition to high bacterial loads and a nutrient-rich environment, hospital effluents in particular may contain concentrations of antibiotics that can facilitate the emergence and spread of resistance genes from one bacterium to another [17]. *Escherichia coli* has been selected as an indicator fecal bacterium that can be exposed to high concentrations of antibiotics in the human or animal gastrointestinal tract and develop resistance before being discharged into sewers and eventually reaching wastewater treatment plants [18]. Fecal bacteria can therefore act as a source of resistance because they can disseminate resistance genes (RGas) to indigenous bacteria [19].

The studies that have addressed the seasonal evolution of antibiotic resistance in HE are rare [19,20]. In general, the amount of medication residues, which is one of the most common sources of AMR in public health, changes according to the seasons in hospital effluents [21,22]. By the correlation between AB consumption and AMR, which are both maximal in winter, AB consumption seems to be the main factor of antibiotic resistance in pathogenic bacteria [21]. In a recent study, it was shown that AMR to meropenems and imipenems is moderate in summer (35%), and low in autumn (18%), whereas it is high in winter (42%) [20]. Indeed, during the colder months of the year, AB prescriptions rise [20] and the abundance of resistant bacteria in HE rises as a result of selection by AB residues in HEs, which are at the base of the winter’s bacterial diseases [23]. As a result, there is a strong link between the rise in AB prescriptions and the increase in ATBs in HE over the winter [20].

The objective of this work is to identify and count antibiotic resistant microorganisms in hospital effluents of 2 units of the Mohamed VI UHC the Mother and Child Hospital and the Ar-razi Surgical Hospital, and to compare the two hospital units in terms of AMR and seasonal variation. A focus has been put on the relationship between the frequency of ARM in HE and the consumption rate of antibiotics inside the hospital.

## 2. Results

### 2.1. Effluent from the ArzH

The Table 1 shows that 15 pathogenic bacterial species were isolated and identified during 24 effluent ArzH tests:6 species are Enterobacteriaceae and represent 40% of the bacterial strains isolated: *Escherichia coli*, *Pantaea agglomerans*, *Hafnea alvii*, *Enterobacter aerogenes*, *Enterobacter cloacae*, and *Klebsiella pneumoniae*;4 are aeromonadacea. and represent 26.66% of the bacterial strains isolated: *Aeromonas caviae, Aeromonas hydrophila, Aeromonas veronii bv sorbia, Aeromonas salmonicidas*;4 are staphylococci and represent 26.66% of the bacterial strains isolated: *Staphylococcus epidermidis, Staphylococc aureus, Staphylococcus huminis*, and *Staphylococcus equorum*.

Seven of the 15 organisms isolated are CRE, which are also resistant to the majority of the 21 ABs tested here, with *Enterobacter aerogenes* having a resistance percentage of 95.4% (Table 1). The percentage of resistance of the 7 CRE strains to the 21 ABs tested was predominant for *Enterobacter aerogenes* (95.24%) followed by *Klebsiella pneumoniae* (90.48%). *Aeromonas veronii bv sorbia* and *Caviae* also showed a high percentage of resistance to all antibiotics (80.95%).

The PEC is a calculated estimate of the concentration of drug residues expected in wastewater based on factors including the amount of the drug consumed over time, the percentage of time it takes the human body to eliminate it through the kidneys and faeces, and the permeability of the wastewater conductors, which is crucial for the onset, development, and persistence of AMR. The PEC is expressed in mg/L as seen in Figure 1 and Figure 3 for ArzH and MCH respectively.

Over the course of the study, the PEC of each of the three carbapenems studied for ArzH varies proportionally with the appropriate AMR: the lowest PEC (0.03 mg/L) is related to the lowest AMR (17.39%), both parameters are related to imipenem; the highest PEC (3.77 mg/L) is related to the highest AMR (43.48%), both parameters are related to meropenem; and the two median values of both parameters are related to ertapenem: PEC (0.05 mg/L) and AMR (26.87%) (Figure 1).

Figure 2 present the seasonal variation of antibiotic resistance AMR in hospital effluent Ar-razi Hospital. The AMR in hospital effluent bacteria for carbapenems and also the 21 other ABs studied varies according to the seasons, indeed it is maximum in winter (84.11%) for the 21ABs and (77.78%) for carbapenems, and during the other seasons the percentage of AMR does not exceed 62% as the maximum value and is detected for the 21 ABs and decreases to 12.70% and is detected for the carbapenems.

When a hospital is involved, as in our situation, the Defined Daily Dosage per 1000 beds (DDD/1000 beds) or per 1000 inhabitants (DDD/1000 inhabitants) is typically used to calculate drug consumption. The consumption of the most popular antibiotic families in ArzH from 2016 to 2019 is shown in Table 2. 

Several families, such as the cephalosporins, carbapenems, and penicillins, are included in the ESBL subgroup. Penicillin use is a major factor in the high consumption of ESBLs, whilst carbapenems are the least popular AB (Table 2).

### 2.2. Effluent from the MCH

The strains isolated and identified in Mother and Child Hospital effluents with their antibiotic resistance are given in Table 3. In the 12 examinations performed, 7 strains were found, of which 58.33% were resistant to carbapenems.

Table 3 shows that the Enterobacteriaceae family dominates in this effluent. Several genera are described: *Klebsiella pneumoniae*: resistant to all carbapenems with an average percentage of resistance of 55.56% and resistance to other ABs tested of 40.03%.*Klebsiella oxytoca:* resistant to meropenem and imipenem-sensitive ertapenem, and resistant to other ABs: 47.61%.*Escherichia coli:* resistant to all 3 carbapenems (44.44%), with high resistance to other ABs: tested: 90.47%.*Serratia odorifera*: resistant to ertapenem (minimum inhibitory concentration: MIC > 0.13), sensitive to the other two carbapenems: meropenem and imipenem; and resistance to other ABs tested: 61.90%.

The Aeromonadacea family is represented by one species: *Aeromonas caviae*: susceptible to carbapenems but with 23.80% resistance to other ABs tested.

Similar to ArzH, the predicted environmental concentrations for each of the three carbapenems under investigation (Figure 3) vary in proportion to the associated AMR. Ertapenem has the lowest PEC (0.145 mg/L), which is correlated with the lowest AMR (28%). While imipenem’s median PEC (1.62 mg/L) equates to a median AMR of 42.86 percent, meropenem has the highest PEC (6.93 mg/L), which corresponds to the greatest AMR (57.14%).

According to the seasons of the year, as depicted in Figure 4, there is a noticeable change in AMR in the MCH effluent. For 21 AB (84.13%) and CBP (77.78%), it reaches its highest levels in the winter. For 21 AB in the autumn, the percentage of AMR does not exceed 61.91%, and for CRB in the summer, it is eliminated.

Table 4 gives the consumption of the main families of antibiotics expressed in Defined Daily Dose per 1000 beds (DDD/1000 beds) in MCH between 2016 and 2019.

In MCH, the most common families of antibiotics are the ESBLs, for which the defined doses per day and per 1000 beds are changing over time with a clear increase between 2016 and 2019.

Penicillin is more commonly consumed at the MCH than ArzH. With the exception of two data for carbapenems in 2016 and 2018, where the consumption in Harz is somewhat higher than in the MCH, Table 2 and Table 4 demonstrate that the daily consumption per bed for the MCH of all ABs is much higher and occasionally double that in ArzH.

### 2.3. Effluents from Both Sites

As shown in Figure 5, Enterobacteria producing carbapenemase are outnumbered by those producing beta-lactamines with extended spectrum (BLES) in both MCH and ArzH. In comparison to Enterobacteria producing carbapenemase, which account for 42.86% and 23.51%, respectively, ESBL-producing bacteria account for 54.28% of MCH and 48.51% of ArzH.

As shown in Table 5, when comparing bacterial resistance to the three carbapenems investigated in this work, there is a peak of resistance to meropenems in the effluents of the MCH with a percentage of resistance of 57.14%, and a minimum of resistance to imipenems in the effluents of the ArzH with a percentage of about 17.39%. The effluents from the two hospitals under study (Mother and Child Hospital and Ar-razi Hospital) on average had 28.23% resistance to the three carbapenems.

## 3. Discussion

### 3.1. Ar-Razi Surgical Hospital

The emergence of PEC in ArzH effluents was also studied in contrast to the evolution of AMR. An ascending evolution of PEC from imipenems (PEC = 0.033 µg/L) passing through ertapenems (PEC = 0.052 µg/L) to meropenems (PEC = 3.77 µg/L) with a parallel increase in AMR: 147.39%, 26.87% and 43.48% respectively, showing a clear relationship between these two parameters (Figure 1).

From the results obtained in ArzH we can conclude that the presence of CRE is associated with resistance to other antibiotics, notably extended-spectrum beta-lactams and 3rd generation cephalosporins and aminoglycosides. Similar results were found in a hospital effluent study in the United States of America [12].

In terms of AMR it is noted in Table 1 that species of the genus Enterobacter present antimicrobial resistance for the three carbapenems studied, with a percentage of 95.24% and 90.48% for the other 21 ABs, respectively in *E*. *aerogenes* and *Klebsielle* spp. This means that this strain is resistant to almost all the available ABs, presenting a MIC higher than 8 g/L, especially for meropenem and imipenem. In a recent study carried out in the effluents of 5 hospitals in Ireland, Enterobacter *cloacae* species resistant to meropenem and ertapenem were also identified [10]. On the other hand, at the Mohamed VI University Hospital in Marrakech [7]. *Enterobacter species* (*E. cloacae* and *E. aerogenes*) were identified in a previous study with additional resistance to imipenem [10], indicating a relative evolution of resistance to carbapenems.

Similarly, 11.1% resistance of E. cloacae species to carbapenems was reported in samples from hospital effluent in Yemen [24].

Strains of the Enterobacteriaceae family, which are gram-negative bacilli (GNBs) such as Enterobacters, Escherichia, Klebsiella, Pantoea, Aeromona, and Hafnia, have been identified in ArzH effluents [25]. All of these GNB are more than 80% resistant to carbapenems and other ESBLs compared to gram-positive bacteria that are coccidia: four *species* of Staphylococus are susceptible to carbapenems and have low resistance to other ESBLs (R = 27%). Therefore, in the ArzH effluent, all gram-negative bacteria have high resistance to carbapenems and other ESBLs, and all gram-positive bacteria are so sensitive.

Species of the Aeromonadaceae family are pathogenic to humans and cause several diseases such as gastroenteritis, skin, soft tissue and muscle infections, and septicemia. Recent studies have revealed high resistance of Aeromonas to carbapenems, especially in Colombia [25,26,27].

The three Aeromonas *species* isolated from ArzH effluent (*A. hydrophila*, *A. caviae*, *A. veronii*) are similar to those identified in several recent studies [25,26,27].

The isolated bacteria found in ArzH effluent represent the majority of bacteria responsible for nosocomial infections, such as enterobacteria and staphylococci, which are abundant during the cold season. According to the results presented in Figure 2, The AMR is higher during winter, when AB consumption is the highest, which leads us to conclude that AB abuse is to be suspected for this increase in AMR during winter for the different strains isolated from ArzH effluents. Indeed, in winter the CRE reaches 77.78% and exceeds 84% while in summer it does not exceed 16.7% and 26.2% for the other AB. The same results were obtained by Schages et al. [19] and Caucci et al. [20] who related them to the abusive consumption of AB in winter following the increase in nosocomial or community contamination, which corroborates the rate of consumption of ABs.

### 3.2. Mother and Child Hospital

From the MCH effluent, we isolated only Negative GRAM bacteria (NGBs), namely *Klebsiella pneumonia*, *Escherichia coli*, *Serratia odorifera*, and *Aeromonas caviae*, which exhibit resistance to carbapenems and ESBLs (Table 3). This antibiotic resistance can be correlated with high consumption of antibiotics such as: penicillins, cephalosporins in general, extended spectrum betalactams, including carbapenems [28].

The evolution of carbapenem AMR is clearly correlated with the PEC, which directly measures drug residues in MCH effluents; it is maximal for meropenem (57.14% AMR for a PEC of 6.93 mg/mL), minimum for ertapenem (28% AMR for a PEC of 0.145 mg/L), and average for miropenems (Figure 3).

The monitoring of the ABR to carbapenems and to the 21 other ABs tested during the different seasons of the year is presented in Figure 5. It is clear that the ABR is maximal in winter: 84.13% for the 21 ABs tested and 77.78% for carbapenems and minimal in spring: 12.70% for the 21 ABs tested and 22.22% for carbapenems. The increase in winter of the frequency of pathogenic diseases related to the decrease of temperatures leads to an abusive consumption of ABs conducting to an increasing of the PEC in the HEs and consequently an automatic raising of the AMR [8,19]

It is noteworthy that we have no official information on seasonal AB use, but when comparing annual consumption, we observed that AB consumption for MCH has increased from 2016 to 2019 (Table 4).

### 3.3. The 2 Sites Effluents

Figure 1 and Figure 3 represent the evolution of PEC of the three carbapenems in comparison with AMR in ArzH and MCH effluents respectively. It can be observed that the increase of AMR correlates well with the increase of PEC in both hospital effluents, which ensures that the overuse of antibiotics induces the progression of AMR [8,19,28].

The PEC and AMR of ArzH follow the following order: Imipenem < Ertapene < Meropenem; on the other hand, the two same parameters of MCH follow the following order: Ertapenem < imipenem < meropenem.

Figure 2 and Figure 4 represent the seasonal evolution of AMR for carbapenems and the 21 other ABs tested in ArzH and MCH effluents. We notice that the high AMR values are attained in winter, which confirms that the abusive use of ABs during this season seems to be one of the main causes of AMR for carbapenems and the 21 other ABs tested [29].

From the results in Table 4, it can be noted that bacteria isolated from MCH effluent are the most resistant to both types of ABs (ESBL and CBP) respectively: 54.28% and 42.86% compared to bacteria isolated from ArzH effluents with resistance percentages: 48.51% for ESBL and 23.51% for CRB (Table 3). These differences in percentages of resistance to CRB and ESBL can be explained by the large difference in annular consumption per bed in AB between the two sites (Table 2 and Table 4), and subsequently by the large PEC difference [30].

Numerous strains from the two effluents studied were isolated: 16 bacterial species out of the 30 microbiological examinations, of which 8 species are carbapenemase-producing Enterobacteriaceae (CEP) and 8 are not CEP representing a high level of resistance to the other ABs studied (21 ABs: ESBL, 2nd and 3rd generation cephalosporins) [31] This resistance exceeds 95%, 90% and 80% respectively for Enterobacter aerogenes, Klebseilla pneumoniae and Aeromonas veronii bv sorbia where Enterobacter aerogenes and Aeromonas veronii bv sorbia are resistant to the three carbapenems studied: ertapenem, imipenem and meropenem with an MIC exceeding 8 µg/L for Enterobacter aerogenes (Table 1 and Table 3). So once the bacteria are Carbapenem-resistant they are automatically resistant to the other ESBLs [30] such MRA is linked to the presence of residues of ABs in hospital effluents and subsequently in the natural environment and further in drinking water [32].

Separately, 12.4% of the isolated strains are resistant to imipenem (Table 1); this percentage is lower than that found by Ory et al. [8], 22.1% (in France); the pollution levels thus vary from one region to another 43.48% are resistant to meropenem and 26.09% are resistant to ertapenem [8].

From the results given in Figure 5, it can be noted that bacteria from MCH are the most resistant to the two types of AB (ESBL and CBP) respectively: 54.28% and 42.86% compared to the resistance of EB from the other site ArzH with percentages of resistance is 48.51%, for ESBL and 23.51% for CRB and ESBL. These differences in percent resistance to CRBs and ESBLs can be explained by the large difference consumption per DDD.1000 bed in AB between the two sites (Table 2 and Table 4). *Klebseilla pneumonia* as carbapenem-resistant enterobacterium (CRE) exceeds 66% resistance to carbapenems (Table 5), lower than the results found by Cahill et al. [10] more than 69% followed by *Escherichia coli* with carbapenemase producing bacteria(CPB) of 50% and *Pantaea agglomerans* and *Aeromonas hydrophila* at the last rank of CPB with resistance of 33% (Table 1) and each which overlaps with previous results [15,30].

Therefore, it can be concluded that hospital effluents could serve as a potential pathway for the transfer of PEC from hospitals to the environment. We have also shown that the two identified species of enterobacteria are resistant even to gentamycin which probably explains the major resistance to this antibiotic due to the presence of these sub-therapeutic concentrations in the studied effluent [33]. From Figure 1, Table 2 and Table 5 we can see that there is a relationship between the rate of PEC of carbapenems in the HEs and CRB. Finally, the emergence and dissemination of AMR in the HEs is largely due to the residues of AB [34,35,36].

The high levels of bacterial resistance found in the HEs were contrasted with those found in a research conducted during the same time period on patients admitted to the University Hospital Mohamed IV in Marrakech.

In the present study 96% of the strains isolated from the HEs are multidrug resistant (Table 1 and Table 3) while this value is about 40% for the study carried out from March 2015 to March 2016 [37]. Therefore, going from the internal environment of the patients to the HE, the AMR of the strains increased by more than 50%; and see the evolution of PEC (Figure 1 and Table 2) we can say that the residues of the antibiotics in the HE help to accentuate the AMR of the pathogenic strains, especially that in our study and the previous study we isolated and identified the same bacterial strains. This does not prevent us from finding AMR values in the HEs that are lower than those found by this same study, but for certain well-defined strains. For example, we identified a strain of *Pseudomonas aerogenusa* that is sensitive to the three carbapenems and has a resistance of more than 28% to the other AMRs (Table 1), whereas the previous study isolated the same strain from hospitalized patients with an AMR to imipenems of 97% and a resistance to ciftazidim of 75%.

The specialities of the MCH and the ArzH are in fact very different. Gynaecology, neonatology, paediatric emergencies, paediatric resuscitation, paediatric surgery, physiotherapy and psychomotricity are services specifically related to MCH functions. Therefore, the variance in ABs consumed in the two hospitals and the specificity of the services provided in each hospital could accelerate the rate of variation in the strains isolated and their AMR.

It is recommended that a policy on the use of ABs be followed which includes

−strengthening surveillance, antibiotic consumption and monitoring of antibiotics that generate bacterial resistance (carbapenems, cephalosporins, fluoroquinolones, amoxicillin-clavulanic acid combination);−control and limitation of the prescription of antibacterial agents which allow savings to be made on last resort antibiotics such as carbapenems, tigecycline, colistin, fosfomycin, daptomycin, linezolid and phenicols) [38];−validation of ABs that can be used in hospital;−the establishment of the list of ABs for controlled distribution and the modalities of this distribution;−setting up and evaluating antibiotic therapy protocols in clinical departments by defining the priority departments in which these protocols must be set up (e.g., emergency department, operating theatre);−conducting prescription audits; −providing information on consumption, costs and new approved ABs.

This investigation reveals a clear association between AB abuse over the winter and the evolution of AMR to different ABs, as illustrated in Figure 2 and Figure 3.

Given the connection between the improper use of ABs and the development of AMR in HEs, it is advised that in the near future, precise techniques like gas or liquid chromatography combined with mass spectroscopy be used to monitor the elements dissolved in these effluents in great detail. AMR at the DNA level of isolated strains should also be carefully and accurately monitored using cutting-edge molecular biology tools. The several genes involved in the development and spread of AMR can be tracked in this way.

## 4. Materials and Methods

### 4.1. Sites of Study

This study was conducted in two health care units of the University Hospital of Marrakech: MCH with a gyneco-obstetric and pediatric vocation with a capacity of 247 beds, and ArzH with a medical and surgical vocation and a capacity of 586 beds.

Emphasis was placed on the microbiological aspect, particularly on the antibiotic resistance of pathogenic bacteria isolated from the effluents of these two hospital units and the seasonal variation of this resistance.

### 4.2. Sampling

Before being discharged through the municipal wastewater system, ArzH and MCH effluents were sampled. Between September 2017 and August 2018, samples were taken separately from each unit’s discharge at a frequency of two samples per month for ArzH (Table 1), and one sample per month for MCH (Table 3).

### 4.3. Strain Identification and ARM Determination

The identification of pathogens and the study of their resistance to antibiotics were performed according to standard Moroccan methods. The membrane filtration technique was considered to detect all the studied germs, such as *Escherichia coli* (I.S.O 9308-1, 2007), Streptococcus and intestinal Enterococci (I.S.O 7899-2; 2007), spores of sulfite-reducing anaerobes (I.S.O 6461-2; 2007), *Staphylococcus aureus*, (N.M 03.7.036; 2012) *Pseudomonas aeruginosa* (I.S.O 16266; 2006) [14].

Selection of staphylococci, fecal streptococci, and pseudomonas was performed by the respective media: Chapman, Slanetz-Bartley, and Cetrimedial agar [39]. The Manual antibiogram was performed using Muller-Hinton agar, which is a non-selective medium [40].

Columbia agar (CNA) medium (with 5% sheep blood) is used to isolate Gram+ (G+) cocci, including staphylococci and streptococci [41].

To isolate Gram-negative bacilli (BGN), a Mac Conkey culture medium is used [41]. These selective and specific media are used by subculturing the colonies to obtain monomorphic cultures. The antibiogram is performed either manually on solid media (halo of inhibition) and by determining the minimum inhibitory concentration (MIC); or with the help of an automated phoenix [42,43] which allows to identify and determine the AMR of isolated germs against a series of antibiotics including carbapenems and others beta-lactams (penicillins, monobactrims, cephalosporins, …) and aminoglycosides, macrolides, phenicolles, cyclines, quinolones, sulfonamides and others. Remember that Caroll et al. [44] demonstrated that the Phoenix system provided adequate performance for the identification and susceptibility testing of 31 frequently encountered *Enterobacteriaceae species*.

The identification and MIC determination are based on colorimetric and fluorescence readings; BD Phoenix AST cassettes are based on a dual indicator system (redox and turbidity) for optimal performance. 

Automated technologies for evaluating antibiotic susceptibility have significantly reduced labor costs and sped up clinician response times. Automation has also significantly improved the dependability of the results due to standardisation of the reagents and procedures and computerization, which removes transcription and categorization errors.

### 4.4. Predicted Environmental Values (PEC)

This parameter concerns the rate of residues of drugs estimated in the environment (here the antibiotics (AB) in HEs) it is calculated based on the rate of consumption of drug (or the AB) in question and also the rate of its renal and fecal elimination during a given period:

PEC _Hospital_: Estimation of drug residues in HE (here the concentration of AB).
PEC _Hospital_ = Q /O × T(1)
and
Q = q × (f + u)(2)

Q: Unchanged quantity of drug (AB) eliminated by the organism.

O: The average flow (here of the HE)

T: Period of time (here one year)

q: quantity consumed in drug (here AB).

f: Percentage of renal elimination of drugs (here AB).

u: percentage of fecal elimination of drugs (here the AB) [32].

## 5. Conclusions

A series of resistant bacteria to carbapenems, notably enterobacteria, was detected in the hospital effluents of the two sites studied at the Mohamed VI Hospital Centre in Marrakech. The presence of AMR in the effluents is mainly related to a high increase in the consumption of antibiotics in these hospitals. This constitutes a danger for the environment knowing that the BLSE represent the last resort of modern world medicine in its fight against pathogenic bacteria.

The presence of antibiotic resistant strains could continue to exchange the genes responsible for this resistance via plasmids with other strains in the environment. Compared to national and international hospital effluents, we deduce that the rate of resistance to carbapenem strains is not negligible despite the low consumption of carbapenem. On the other hand, the abusive use of antibiotics during the period of the appearance of COVID 19, we propose a continuation of this study that will concern the evolution of the AMR during this pandemic. This could cause health problems and environmental impacts in the near future. In view of all these elements, it is suggested that these effluents have to be treated before being discharged into the municipal network. 

## Figures and Tables

**Figure 1 antibiotics-11-01103-f001:**
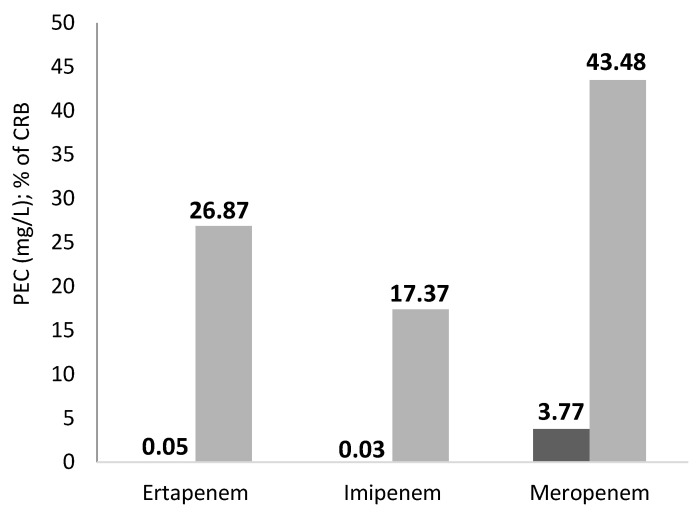
Graphical representation of the Predicted Environmental Concentration (■) (in mg/L): in comparison with the percentage of Carbapenem-resistant bacteria (■) (in %) in ArzH.

**Figure 2 antibiotics-11-01103-f002:**
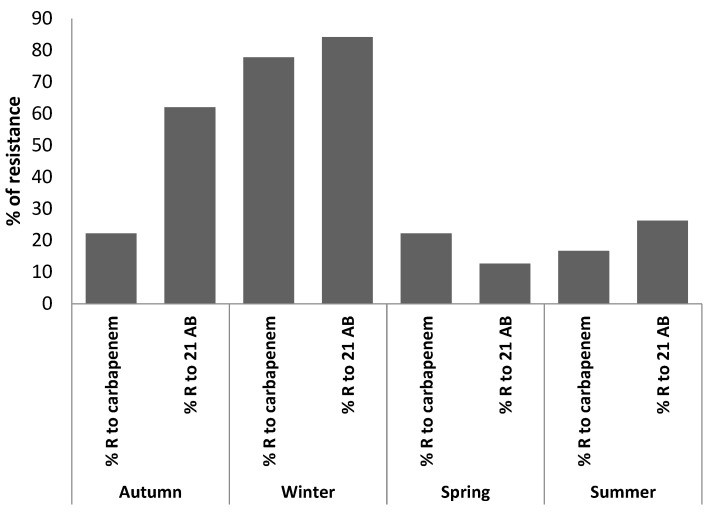
Seasonal variation of antibiotic resistance AMR in Ar-razi Hospital effluent.

**Figure 3 antibiotics-11-01103-f003:**
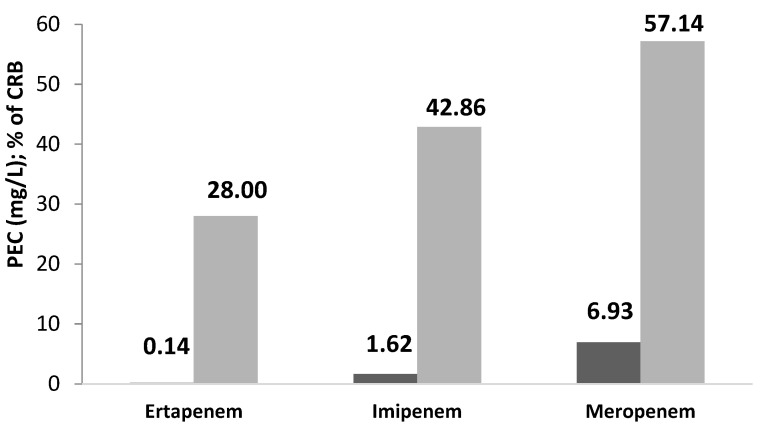
Graphical representation of the Predicted Environmental Concentration (PEC) (■) (in mg/L): in comparison with the percentage of Carbapenem-resistant bacteria CRB) (■) (in %) in MCH.

**Figure 4 antibiotics-11-01103-f004:**
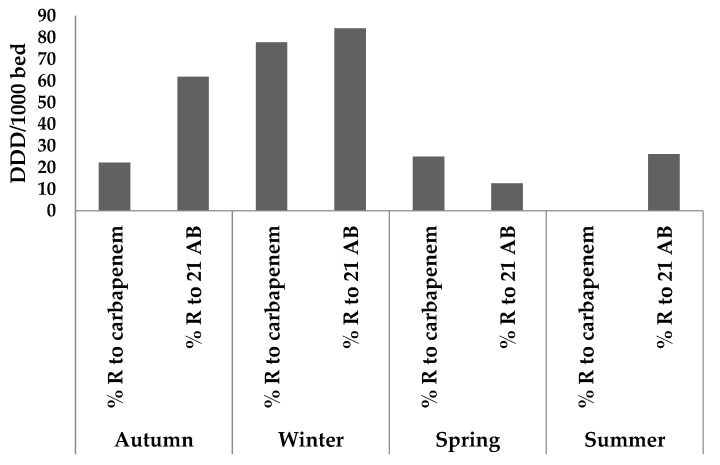
Seasonal variation in the percentage of antimicrobial resistance in MCH effluents.

**Figure 5 antibiotics-11-01103-f005:**
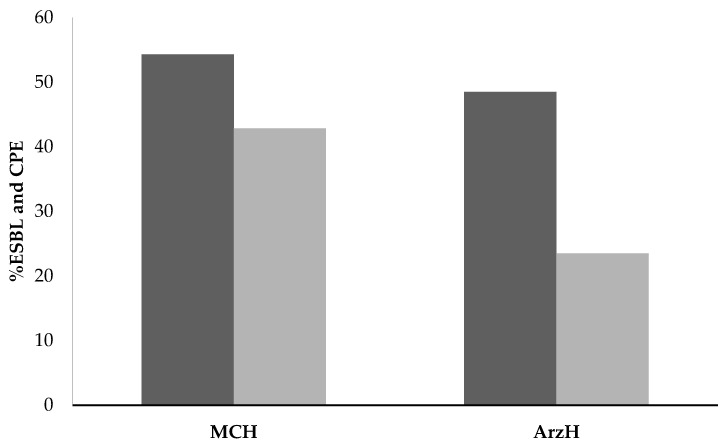
Percentage of extended-spectrum betalatamase-producing enterobacteria (ESBL) (■) and Carbapenemase-producing Enterobacteria (CPE) (■) in ArzH, and MCH.

**Table 1 antibiotics-11-01103-t001:** The strains isolated and identified in Ar-razi Hospital effluent with their antibiotic susceptibility tests results.

Exam No.	Collection Season	Identified Germs	Ertapenem		Imipenem		Meropenem		% R to Carbp	Frequency of Bacterial Resistance among 21 AB	
			R/S	MIC	R/S	MIC	R/S	MIC		R/N.AB	%
1	Autumn	*Escherichia coli*	S		S	<0.25	R	>8	0	17/21	80.95
2	Autumn	*Escherichia coli*	S		S	0.25	R	>8	33.33	10/21	47.62
3	Autumn	*Escherichia coli*	S	0.25	S	<0.25	R	>8	33.33	16/21	76.19
4	Autumn	*Pantaea agglomerans*	R	>1	S		S	=0.5	33.33	9/21	42.86
5	Autumn	*Hafnea alvii*	S		S		S		0	9/21	42.86
6	Autumn	*Aeromonas caviae*	S	<0.13	S	<0.25	R	>8	33.33	17/21	80.95
7	Winter	*Escherichia coli*	R	0.25	S		R	>8	66.66	18/21	85.71
8	Winter	*Enterobacter aerogenes*	R	>1	R	>8	R	>8	100	20/21	95.24
9	Winter	*Klebsiella pneumoniae*	R	>1	I	1	R	>8	66.66	19/21	90.48
10	Winter	*Enterobacter aerogenes*	R	>1	R	>2	R	>8	100	19/21	90.48
11	Winter	*Aeromonas hydrophila*	S		S		R	>8	33.33	13/21	61.91
12	Winter	*Aeromonas veronii bv sorbia*	R	>1	R	>2	R	>8	100	17/21	80.95
13	Spring	*Escherichia coli*	S	<0.13	S	0.25	S	<0.13	0	00/21	00
14	Spring	*Escherichia coli*	S	0.25	S	0.25	S		0	01/21	4.76
15	Spring	*Aeromonas caviae*	S		S	<0.25	S		0	4/21	19.05
16	Spring	*Aeromonas caviae*	S		S	<0.13	S	<0.25	0	2/21	9.52
17	Spring	*Staphylococcus epidermidis*	S		S		S		0	3/21	14.29
18	Spring	*Pseudomonas aeroginosa*	S		S		S		0	6/21	28.57
19	Summer	*Aeromonas salmonicidas*	S		S		S		0	7/21	33.33
20	Summer	*Staphylococcus aureus*	S		S		S		0	4/21	19.05
21	Summer	*Staphylococcus huminis*	S		S		S		0	10/21	47.62
22	Summer	*Staphylococcus equorum*	S		S		S		0	6/21	28.57
23	Summer	*Escherichia coli*	S		S		S		0	3/21	14.29
24	Summer	*Enterobacter cloacae*	S	<0.13	S	<0.25	S		0	3/21	14.29

R/S: resistant/sensitive. I: indifferent. MIC: Minimum Inhibitory Concentration (in mg/L). R/N.AB: number of resistant strains/number of antibiotics studied.

**Table 2 antibiotics-11-01103-t002:** Consumption of the main families of antibiotics expressed in Defined Daily Dose per 1000 beds (DDD/1000 beds) in ArzH between 2016 and 2019.

Year	2016	2017	2018	2019	Standard Deviation
ESBL	313.58	367.25	280.72	420.96	48.48
Penicillins	197.28	251.61	120.87	221.90	38.84
Cephalosporins	104.69	111.45	147.29	180.45	27.90
Carbapenems	11.61	4.18	12.55	18.61	3.84

(ESBL): Extended Spectrum Beta-Lactam.

**Table 3 antibiotics-11-01103-t003:** The strains isolated and identified in Mother and Child Hospital effluents with their antibiotic resistance.

Exam No.	Collection Season	Identified Germs	Ertapenem		Imipenem		Meropenem		% R to Carbp	Frequency of Bacterial Resistance among 21 AB	
			R/S	MIC	R/S	MIC	R/S	MIC		R/N.AB	% Resistance
1	Autumn	*Klebsiella oxytoca*	S		S		S		0	10/21	47.61
2	Autumn	*Escherichia coli*	S		S		S		0	6/21	28.57
3	Autumn	*Klebsiella pneumoniae*	S		R	>4	R	>2	66.66	12/21	57.14
4	Winter	*Klebsiella pneumoniae*	R	>0.13	R	>0.25	R	>0.13	100	14/21	66.66
5	Winter	*Escherichia coli*	S		R	>2	R	>1	66.66	19/21	90.47
6	Winter	*Serratia odorifera*	R	>0.13	S		S		33.33	13/21	61.90
7	Spring	*Klebsiella pneumoniae*	S		S	<0.25	S	<0.13	0	3/21	14.28
8	Spring	*Escherichia coli*	R		S		S		33.33	3/21	14.28
9	Spring	*Escherichia coli*	S		S		R		33.33	6/21	28.57
10	Summer	*Klebsiella oxytoca*	R		S	<0.25	R	>1	66.66	10/21	47.61
11	Summer	*Aeromonas caviae*	S		S	<0.13	S		0	5/21	23.80
12	Summer	*Escherichia coli*	S		S		S		0	4/21	19.00

R/S: resistant/sensitive. I: indifferent. MIC: Minimum Inhibitory Concentration. R/N.AB: number of resistant strains/number of antibiotics studied.

**Table 4 antibiotics-11-01103-t004:** Consumption of the main families of antibiotics expressed in Defined Daily Dose per 1000 beds (DDD/1000 Beds) in MCH between 2016 and 2019.

Year	2016	2017	2018	2019	Standard Deviation
ESBL	539.39	611.71	651.91	821.02	82.51
Penicillins	357.99	467.68	459.50	577.65	56.97
Cephalosporins	170.32	138.70	181.50	219.29	22.94
Carbapenems	11.12	5.32	10.91	24.08	5.61

(ESBL): Extended Spectrum Beta-Lactam.

**Table 5 antibiotics-11-01103-t005:** Percentage of resistance to the three carbapenems in the two sites ArzH and MCH.

Study Site	% of Ertapenem-Resistant Bacteria	% of Imipenem-Resistant Bacteria	% of Meropenem-Resistant Bacteria	% Resistance to the All Carbapenem Family
ArzH	26.87	17.39	43.48	23.51
MCH	28.57	42.86	57.14	42.86
Average	18.48	38.42	33.54	28.23

## Data Availability

The data is contained in the article.

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
