# Peer review of "Microbial Resistance to Carbapenems in Effluents from Gynaecological, Paediatric and Surgical Hospital Units"

_antibiotics, 2022, doi:10.3390/antibiotics11081103_

Round 1

Author Response

Manuscript ID: antibiotics-1796576

Response to reviewer

Dear Reviewer,

We would like to thank you for your time and effort in providing feedback on our manuscript and are grateful for the insightful comments and valuable improvements to our paper. The manuscript is corrected and according to the comments point by point. All the modifications performed on the manuscript as well as our responses below are in blue.

1) The paper uses many abbreviations which confuses the readers.

Author response:  The corresponding change has been done.

2) Secondly, the abbreviations are not spelt out in their original complete form in the beginning of the paper, to assist authors. Eg- The abstract mentions AB (Line 18) and PEC (Line 19) which are spelt out only later to us readers of what they mean.

Author response:  The corresponding change has been done.

3) The Y axis are not labeled on any of the graphs.

Author response:  The corresponding change has been done.

4) Based on the data and conclusions, there is a difference in the type of AMR bacterial isolates from the two hospital units – MCH and ArzH, and the authors indicate it is dependent on the type of antibiotics used in those hospital units. Could there also be a relation between one hospital unit related to childcare and other not? It would be nice to have that factored in as well in the paper.

Author response: The specialities of the MCH and the ArzH are in fact very different. Gynaecology, neonatology, paediatric emergencies, paediatric resuscitation, paediatric surgery, physiotherapy and psychomotricity are services specifically related to MCH functions. Therefore, the variance in ABs consumed in the two hospitals and the specificity of the services provided in each hospital could accelerate the rate of variation in the strains isolated and their AMR.

5) The authors do not suggest many measures to be taken to monitor these antibiotic elements in the effluents. Additionally, they suggest using cutting-edge molecular biology tools to determine AMR at the DNA level (Line 404, 405). This is very discrete information and requires more clarification using references and additional explanation. Thirdly, there are no comments made in the paper regarding the regulatory use and release of these effluents in these hospital units. What better can be done ?

Author response: It is recommended that a policy on the use of ABs be followed which includes

- strengthening surveillance, antibiotic consumption and monitoring of antibiotics that generate bacterial resistance (carbapenems, cephalosporins, fluoroquinolones, amoxicillin-clavulanic acid combination);

- control and limitation of the prescription of antibacterial agents which allow savings to be made on last resort antibiotics such as carbapenems, tigecycline, colistin, fosfomycin, daptomycin, linezolid and phenicols) [38];

- validation of ABs that can be used in hospital;

- the establishment of the list of ABs for controlled distribution and the modalities of this distribution;

- setting up and evaluating antibiotic therapy protocols in clinical departments by defining the priority departments in which these protocols must be set up (e.g. emergency department, operating theatre);

- conducting prescription audits;

- providing information on consumption, costs and new approved ABs.

Reference :       38        Gandolière, A. (2015). Évaluation de la politique de bon usage des                 antibiotiques du CHR Metz-Thionville de 2007 à 2014: confrontation                     au suivi des consommations d’antibiotiques et des résistances                            bactériennes (Doctoral dissertation, Université de Lorraine).

6) In Line 68-69, authors suggest carbapenem deactivating enzymes have an efficient dissemination. How is this statement made, can this please be explained in detail?

Author response: It has been indicated that there is a dissemination of the genes encoding the enzymes and not a direct spread of the enzymes. Bacteria cannot naturally synthesise the carbapenem hydrolysing enzymes, but once they receive the genes responsible for synthesising these enzymes, they can synthesise them and then become resistant.

7) The authors only employ old methods of strain identification and determination. Using next gen sequencing tool could add to the value of the paper.

Author response: In order to identify strains more precisely and calculate their AMR, we used the automated BD Phoenix system in conjunction with the conventional manual identification technique utilising selective media of various degrees. Remember that Caroll et al.(2006) demonstrated that the Phoenix system provided adequate performance for the identification and susceptibility testing of 31 frequently encountered Enterobacteriaceae species.

Reference :       Carroll, K. C., Glanz, B. D., Borek, A. P., Burger, C., Bhally, H. S., Henciak,                       S., & Flayhart, D. (2006). Evaluation of the BD Phoenix automated                            microbiology system for identification and antimicrobial susceptibility testing                     of Enterobacteriaceae. Journal of clinical microbiology, 44(10), 3506-3509.

8) The Line 438 – automated phoenix should be explained using 2-3 lines instead of just giving the reference.

Author response: The automated BD Phoenix system is used to identify bacterial isolates and determine their sensitivity to a series of ABs.

The identification and MIC determination are based on colorimetric and fluorescence readings; BD Phoenix AST cassettes are based on a dual indicator system (redox and turbidity) for optimal performance.

Automated technologies for evaluating antibiotic susceptibility have significantly reduced labour costs and sped up clinician response times. Automation has also significantly improved the dependability of the results due to standardisation of the reagents and procedures and computerization, which removes transcription and categorization errors.

Please accept, Excellency, the assurances of my most respectful greetings.
With my Best Regards

El hassan Loumame

hloumame@gmail.com

Reviewer 2 Report

It is not clear whether the novelty of experimental approach used by authors. It will help to know if the approach used here has been used earlier and if authors have improved it,  Authors can improve their manuscript by consistent use of italics for bacterial species and reporting % values up to 1 decimal point.  In addition, authors need to check and update abbreviations at several positions in introductions. Either they were never expanded or used only once so no abbreviation is needed. Detailed comments are following:

Line 64 - CRE, strains = Is , a typo as its not needed here.

Line 111 - Please complete sentence by informing what are you correlating.

Lines - 119-121 = Please explain why these 2 hospitals and which 2 units were selected. It will be great if authors could assess how their study on these hospitals/units could be applicable to medical facilities around the globe.

 Line 135 - Staphylococc, = Please remove comma and complete the name of bacteria.

Line 147 - Please mention the units of MIC.

Lines 183-187 = Authors are mentioning values from their table which is not needed in text here. It could help to rather explain or speculate why different class of antibiotics were used at different concentrations and the reasons behind their increase/decrease for the period 2016-2019.

Lines 232-247 = Same comment about redundancy as made earlier for ArzH. Here authors could compare MCH and ArzH  for their trends.

Line 438 - Please mention "The autoradiogram" instead of which

Lines 467-468 = Sentences needs to be reframed for clarity.

Author Response

Manuscript ID: antibiotics-1796576

Response to reviewer

Dear Reviewer,

We would like to thank you for your time and effort in providing feedback on our manuscript and are grateful for the insightful comments and valuable improvements to our paper. The manuscript is corrected and according to the comments point by point. All the modifications performed on the manuscript as well as our responses below are in blue.

Reviewer questions

It is not clear whether the novelty of experimental approach used by authors. It will help to know if the approach used here has been used earlier and if authors have improved it.

Author response: The BD phenix method is commonly applied to the analysis of blood, urine and other biological fluids. The originality of our experimental approach lies in the application of this method to hospital effluents. The frequent use of this method will certainly lead to its improvement in the long run.

Authors can improve their manuscript by consistent use of italics for bacterial species and reporting % values up to 1 decimal point. In addition, authors need to check and update abbreviations at several positions in introductions. Either they were never expanded or used only once so no abbreviation is needed.

Author response: The corresponding changes have been done

Detailed comments are following:

1) Line 64 - CRE, strains = Is , a typo as its not needed here.

Author response: The corresponding change has been done

2) Line 111 - Please complete sentence by informing what are you correlating.

Author response: The corresponding change has been done

3) Lines - 119-121 = Please explain why these 2 hospitals and which 2 units were selected. It will be great if authors could assess how their study on these hospitals/units could be applicable to medical facilities around the globe.

Author response: The current work is an extension of previous scientific research carried out by El-Ogri Fouzia in 2015 and Qadori Asmaa in 2016 on the hospital units of the University Hospital of Marrakech. These findings, along with our own, which included a physico-chemical and microbiological characterization study, will then serve as the foundation for a treatment system proposal for the hospital effluents under study that may be applicable to hospitals of the same size anywhere in the world. 

4) Line 135 - Staphylococc, = Please remove comma and complete the name of bacteria.

Author response: The corresponding change has been done.

5) Line 147 - Please mention the units of MIC (mg/L)

Author response: The corresponding change has been done.

6) Lines 183-187 = Authors are mentioning values from their table which is not needed in text here. It could help to rather explain or speculate why different class of antibiotics were used at different concentrations and the reasons behind their increase/decrease for the period 2016-2019.

Author response: The corresponding change has been done.

7). Lines 232-247 = Same comment about redundancy as made earlier for ArzH. Here authors could compare MCH and ArzH  for their trends.

Author response: The corresponding change has been done.

8) Line 438 - Please mention "The autoradiogram" instead of which selection of staphylococci, fecal streptococci, and pseudomonas was performed by the respective media: Chapman, Slanetz-Bartley, and Cetrimedial agar [40]. The Manual antibiogram was performed using Muller-Hinton agar, the antibiogram is a non-selective medium [41].

Author response: The corresponding change has been done.

9) Lines 467-468 = Sentences needs to be reframed for clarity.

Author response: The corresponding change has been done.

Please accept, Excellency, the assurances of my most respectful greetings.
With my Best Regards

El hassan Loumame

hloumame@gmail.com

Round 2

Reviewer 1 Report

It is good to go now! All corrections seem fine.